# Role of Magnesium in Skeletal Muscle Health and Neuromuscular Diseases: A Scoping Review

**DOI:** 10.3390/ijms252011220

**Published:** 2024-10-18

**Authors:** Sara Liguori, Antimo Moretti, Marco Paoletta, Francesca Gimigliano, Giovanni Iolascon

**Affiliations:** 1Department of Medical and Surgical Specialties and Dentistry, University of Campania “Luigi Vanvitelli”, Via De Crecchio n. 4, 80138 Naples, Italy; sara.liguori@unicampania.it (S.L.); marco.paoletta@unicampania.it (M.P.); giovanni.iolascon@unicampania.it (G.I.); 2Department of Mental and Physical Health and Preventive Medicine, University of Campania “Luigi Vanvitelli”, Largo Madonna delle Grazie n. 1, 80138 Naples, Italy; francesca.gimigliano@unicampania.it

**Keywords:** magnesium, skeletal muscle, neuromuscular disease, muscle strength, dietary supplement

## Abstract

Magnesium (Mg) is a vital element for various metabolic and physiological functions in the human body, including its crucial role in skeletal muscle health. Hypomagnesaemia is frequently reported in many muscle diseases, and it also seems to contribute to the pathogenesis of skeletal muscle impairment in patients with neuromuscular diseases. The aim of this scoping review is to analyze the role of Mg in skeletal muscle, particularly its biological effects on muscle tissue in neuromuscular diseases (NMDs) in terms of biological effects and clinical implications. This scoping review followed the PRISMA-ScR (Preferred Reporting Items for Systematic Reviews and Meta-Analyses Extension for Scoping Reviews) guidelines. From the 305 studies identified, 20 studies were included: 4 preclinical and 16 clinical studies. Preclinical research has demonstrated that Mg plays a critical role in modulating pathways affecting skeletal muscle homeostasis and oxidative stress in muscles. Clinical studies have shown that Mg supplementation can improve muscle mass, respiratory muscle strength, and exercise recovery and reduce muscle soreness and inflammation in athletes and patients with various conditions. Despite the significant role of Mg in muscle health, there is a lack of research on Mg supplementation in NMDs. Given the potential similarities in pathogenic mechanisms between NMDs and Mg deficiency, further studies on the effects of Mg supplementation in NMDs are warranted. Overall, maintaining optimal Mg levels through dietary intake or supplementation may have important implications for improving muscle health and function, particularly in conditions associated with muscle weakness and atrophy.

## 1. Introduction

Magnesium (Mg), an alkaline earth metal and the eighth most abundant element on Earth, is essential for various biochemical and physiological processes [1]. It constitutes approximately 25 g in the human body, primarily located in bones (over half) and muscles/soft tissues (one-third), with intracellular levels significantly exceeding those in extracellular fluid [2,3,4]. Mg is critical for bone health, as it influences the formation of hydroxyapatite crystals, thereby preventing them from becoming excessively large or brittle [1]. Magnesium plays a crucial role in cellular metabolism by acting as a cofactor for over 300 enzymes and being indispensable in ATP metabolism, thereby contributing to both aerobic and anaerobic energy generation and glycolysis [5]. This electrolyte is also involved in sodium/potassium ATPase activity, maintaining intracellular potassium, and acts as a physiological calcium channel blocker [6]. The human body contains about 25 g of Mg in adulthood, with just over half of this located in bones, and a further third in muscles and soft tissues. The intracellular concentration is about ten times that of the extracellular fluid [5,6]. Intracellular Mg is primarily stored in the mitochondria, contributing to ATP (adenosine triphosphate) synthesis from ADP (adenosine diphosphate) and inorganic phosphate. As part of the Mg-ATP complex, it yields the bioactive form of ATP. Moreover, Mg promotes the coupled state necessary for mitochondrial oxidative phosphorylation and helps avoid the production of oxygen-derived free radicals in mitochondria [7]. Magnesium is essential for DNA and RNA synthesis, supplying adequate purine and pyrimidine nucleotides, and its presence is required for the activation of the adenylate cyclase, involved in the regulation of cellular activity [8]. Dietary sources of magnesium include vegetables (e.g., spinach), legumes, nuts, seeds, animal products, and water. However, the magnesium content in plant foods has declined due to agricultural practices like lime application to acid soils [3,9]. Magnesium, like calcium, is absorbed in the duodenum and ileum through active and passive processes. The kidney also plays a central role in magnesium homeostasis through active reabsorption influenced by the sodium load in the tubules and the acid–base balance [10]. A high dietary intake of calcium (approximately 2600 mg/day) with a high sodium intake enhances magnesium excretion [11]. On average, magnesium intake is less than current recommendations by about a third in women and a quarter in men [12]. According to the Food and Nutrition Board (FNB) at the Institute of Medicine of the National Academies, the intake recommendations for magnesium in adults range from 410 to 420 mg for males and 320 to 360 mg for females [13]. Primary nutritional magnesium deficiency is rarely observed in humans unless a low intake is accompanied by an excessive loss, such as during prolonged diarrhea. Although most of the early signs of deficiency are neurological, hypomagnesemia increases intracellular calcium, resulting in muscle cramps, hypertension, and vasospasms [1,2]. Moreover, Mg deficiency is related to the development of a reversible, metabolic cardiomyopathy [14]. Hypomagnesemia initially presents with weakness, loss of appetite, fatigue, nausea and vomiting. Subsequently, it can be complicated by muscle spasms and cramps, dysesthesia, cardiovascular manifestations, convulsions, cognitive impairment, and, in severe deficiency cases, hypocalcemia or hypokalemia [15]. Many muscle diseases, including sarcopenia, inflammatory muscle diseases, and neuromuscular disorders (NMDs), can be accompanied by significant hypomagnesemia [16,17]. The scientific literature does not agree on the significance of this association, as anatomical and/or functional damage to muscle can be primary and cause hypomagnesemia or be secondary to a previous chronic magnesium deficiency. This uncertainty hinders the ability to develop targeted interventions such as in the case of NMDs, potentially leading to inadequate responses and missed nutritional guidance for patients. It should be emphasized that chronic inflammation and oxidative stress can also be considered causes or effects of magnesium deficiency since this electrolyte plays an essential role in regulating the pathways underlying these conditions [18,19]. This scoping review focuses on the role of Mg in skeletal muscle, particularly its biological effects on muscle tissue in NMDs and its clinical and therapeutic implications.

## 2. Materials and Methods

This scoping review followed the PRISMA-ScR (Preferred Reporting Items for Systematic Reviews and Meta-Analyses Extension for Scoping Reviews) guidelines [20].

The first step was the creation of a technical expert panel (TEP) consisting of 5 medical specialists with expertise in skeletal muscle disorders and confidence with scoping review methodology.

### 2.1. Search Strategy

The TEP conducted a search on PubMed (Public MedLine, run by the National Center of Biotechnology Information (NCBI) of the National Library of Medicine Bethesda, Bethesda, MD, USA) using the following MeSH (Medical Subject Heading) terms: “Magnesium” or “Magnesium Compounds” and “Muscular Dystrophy, Duchenne” or “Becker dystrophy” or “Myasthenia Gravis” or “Charcot-Marie-Tooth Disease” or “Muscular Atrophy, Spinal” or “Glycogen Storage Disease Type II” or “neuromuscular diseases” or “Muscle, skeletal” (Table 1). The choice of these NMDs included for the analysis is based on the main neuromuscular disorder per lesion level.

### 2.2. Study Selection

The TEP defined the characteristics of the sources of evidence, considering articles published from inception to 31 May 2024, including only those written in English (see Table 2 for further details about eligibility criteria).

### 2.3. Data Extraction and Quality Assessment

All data extracted from the included studies were qualitatively analyzed. The study selection and data extraction were performed independently by two authors (SL and MP), and in the case of any controversies, a third author (AM) was consulted.

## 3. Results

The study selection process is reported in Figure 1. We screened 305 articles from the PubMed database. Based on titles and abstracts, and following our selection criteria, a total of 275 papers were excluded. After reading the full texts, a further 10 articles were excluded. The remaining 20 articles (published up to May 2024) met the inclusion criteria and are described in Table 3 [21,22,23,24,25,26] and Table 4 [27,28,29,30,31,32,33,34,35,36,37,38,39,40].

## 4. Discussion

The findings from preclinical and clinical studies underscore the critical role of Mg in skeletal muscle function, as outlined in Figure 2.

### 4.1. Magnesium in Skeletal Muscle Metabolism

Magnesium is crucial in regulating glucose, lipid, and protein metabolism, which may significantly influence the muscle–fat tissue cross-talk.

Indeed, Mg supplementation might counteract intramuscular fat infiltration and fat content, as demonstrated in experimental models of CS-induced muscle atrophy; this supports magnesium’s potential role in regulating systemic lipid metabolism in muscle and bone marrow [17]. Furthermore, magnesium supplementation may inhibit muscle proteolysis by counteracting the calcium-dependent proteolytic system, which includes cysteine proteases known as calpains [22]. Conversely, low extracellular magnesium levels, often resulting from inadequate dietary intake, can negatively impact glucose metabolism by reducing glucose uptake in myotubes and decreasing the activity of glyceraldehyde-3-phosphate dehydrogenase (GAPDH) [26]. This alteration in glucose metabolism can lead to different metabolic patterns in pyruvic acid and carnosine levels [25].

Another important role of Mg is its influence on oxidative stress markers related to skeletal muscle health. According to Liu et al., nutritional intake of Mg seems to regulate the balance between free radicals and anti-oxidant production, with benefits for muscle tissue [21]. However, the positive effects of magnesium supplementation may be diminished in individuals with hypokinesia, including those with sedentary lifestyles [29,30]. This reduction is likely due to changes in magnesium metabolism-related tissues, such as the depletion of glycogen stores, inadequate anaerobic glycolysis, and a decrease in mitochondrial function and number [30]. Consequently, these detrimental effects on cellular metabolism could create a vicious cycle that impacts ATP production, the activity of Na+, K+-ATPase, oxidative metabolism, and cell membrane permeability. It should be underlined that Mg supplementation in people with normal levels of this cation does not further increase serum Mg and does not improve neuromuscular activity, muscle-related symptoms, or exercise performance, as demonstrated in athletes supplemented with daily capsules of Mg oxide [28].

Therefore, the utility of magnesium supplementation in young athletes has been questioned, particularly since serum magnesium levels typically rise post-exercise due to its release from damaged muscles [41]. Finally, some of the clinical studies included Mg supple-mentation in conjunction with other nutritional supplements, such as vitamin C and B6. This concurrent supplementation can introduce confounding variables that obscure the specific contributions of Mg to outcomes related to oxidative stress and other biological processes.

On the other hand, Mg supplementation might have a rationale for use in athletes to promote skeletal muscle function recovery after exercise-induced damage, considering its anti-inflammatory properties, thus hampering exercise-induced fatigue and muscle soreness and fostering post-exercise recovery [34,35,36,37,42]. As reported by Tarsitano et al., increasing magnesium intake by 10–20% above the recommended dose, especially through supplements taken 2 h before exercise, may be beneficial for active individuals [43]. However, Wang et al. do not support the benefits of magnesium for muscle fitness in most athletes and physically active individuals who already have relatively high magnesium levels [44].

Given these insights, Mg supplementation is advisable for individuals, particularly those at risk of muscle atrophy such as those with NMDs, including those receiving chronic CSs (e.g., Duchenne Muscular Dystrophy), as part of a comprehensive nutritional and exercise strategy.

### 4.2. Magnesium and Muscle Fiber Transition

Magnesium supplementation plays a role in counteracting muscle atrophy, which progressively modifies muscle tissue quality and fiber type, including in conditions such as sarcopenia. In sarcopenia, a hallmark of the disease is the tendency for a fast-to-slow twitch fiber transition compared to other forms of muscle atrophy [45]. The combination of Mg and low-magnitude, high-frequency vibration (LMHVF) may increase muscle mass and fiber CSA while also preventing fiber-type conversion in favor of type II fiber [24]. Additionally, Mg supplementation has been shown to significantly improve muscle mass, strength, and performance [39].

The structural and clinical benefits of Mg supplementation in skeletal muscle function appear to be attributable to the modulation of the IGF-1/PI3K/Akt pathway. Mg, both with and without LMHVF, has been shown to increase the expression of mTOR and pAkt [24], which are key regulators of anabolic and catabolic signaling of skeletal muscle, thereby preventing muscle atrophy.

Thus, Mg supplementation has biological plausibility as a therapeutic option for managing muscle atrophy and muscle weakness in various conditions, including sarcopenia [45].

### 4.3. Magnesium and Skeletal Muscle Regeneration

Considering that normal levels of Mg seem essential to ensure the regenerative capacity of skeletal muscle fibers, Mg supplementation promotes myogenic differentiation, enhancing muscle regeneration through mTOR signaling and the subsequent activation of key myogenic genes, such as Myf5, Myod, and Myog [23], in satellite cells (SCs), which are responsible for muscle repair and regeneration.

On the contrary, low serum Mg, promoting the downregulation of Myog, MyHC, and Myomixer and reducing autophagic flux, adversely affects the muscle fusion process, with consequently thinner myotubes [25,26]. However, in C2C12 cells, low and high extracellular Mg concentrations induce oxidative stress and inhibit myoblast fusion, affecting myogenesis [26].

It could be hypothesized that Mg could also play a role in the differentiation and activity of SCs in pathological conditions (e.g., sarcopenia, NMDs), where SC alterations are typically observed.

### 4.4. Analgesic Effects of Magnesium in Muscle Disease

It has been acknowledged that Mg does not possess direct analgesic properties. However, it plays a significant role in preventing central sensitization—a condition in which the nociceptive pathways of the central nervous system become excessively sensitive due to repeated pain signals from peripheral injuries. By blocking NMDA receptors and inhibiting the influx of calcium ions into cells, magnesium offers pain relief, thereby confirming its role as a calcium channel blocker [46].

Magnesium intake also seems to relieve muscle soreness, resulting from exercise-induced muscle damage, decreasing lactate levels. Magnesium is also recognized as being an analgesic for its muscle relaxant and vasodilator properties, being used in patients with myofascial pain syndrome, supporting the pleiotropic action of this cation on skeletal muscle [38]. Its efficacy highlights magnesium’s multifaceted role for individuals experiencing persistent muscle pain, particularly if traditional pain relief methods are insufficient.

### 4.5. Magnesium Supplementation on Muscle Health Across Various Disorders: A Focus on NMDs

Magnesium is well known for its role in muscle health, and its supplementation, when appropriate, is an effective strategy for improving muscle performance. In conditions characterized by Mg depletion, such as alcoholic liver disease, Mg supplementation significantly increases not only serum Mg levels and tissue content but also muscle mass and strength [32]. Similarly, Mg supplementation may improve respiratory muscle strength and clinical outcomes in patients with cystic fibrosis (CF) [33]. Additionally, it has anti-inflammatory effects, enhancing skeletal muscle mass, strength, and quality of life in individuals with chronic obstructive pulmonary disease (COPD) [37].

Current research also explores this cation in various neurological diseases, such as stroke, epilepsy, Alzheimer’s disease, and Parkinson’s disease [16]. However, we did not find any study about Mg supplementation in NMDs. This is surprising considering that NMDs and Mg depletion may share certain pathogenetic mechanisms. For instance, in DMD, the disruption of the dystrophin–glycoprotein complex (DGC), which is closely linked to several ion channels, contributes to abnormal ion activity. This leads to decreased total Mg, P, and Zn while increasing total Ca and Na levels [19]. Specifically, free Mg muscle content is reported to be lower in DMD patients, as shown by phosphorus nuclear magnetic resonance spectroscopy (31P NMRS) in muscle biopsies [47]. This ion dysregulation may contribute to the progressive muscle damage characterized by fat and fibrotic tissue accumulation, inflammation, and metabolic changes observed in DMD [48]. Furthermore, increased ROS-induced protein damage and lipid peroxidation are seen in muscle biopsies of DMD patients [48]. Given that hypomagnesemia is frequently observed in NMDs, maintaining Mg homeostasis through proper dietary intake seems advisable.

Recently, the role of satellite cells (SCs) has gained attention in NMD research [49], particularly in DMD, where these cells, despite being numerous, often exhibit dysfunction when undergoing apoptosis or senescence due to cell cycle arrest and impaired autophagic mechanisms [49]. Senescent cells increase the release of growth factors and pro-inflammatory cytokines, compromising the function of neighboring cells. Similar events are observed in sarcopenia, although in this condition, a reduction in the number of satellite cells prevails over their dysfunction [50].

Given magnesium’s modulatory role in the transcription of myogenic genes, which affects SC differentiation, function, and cell senescence, normalizing magnesium levels could promote muscle regeneration in NMDs.

### 4.6. Exploring the Impact of Magnesium Intake in Clinical Practice: Key Insights and Lessons Learned

Based on findings from both preclinical and clinical studies regarding the role of Mg in skeletal muscle, several clinical implications can be derived:Nutritional recommendations for muscle health by addressing Mg values: Nutritional counseling to ensure adequate dietary magnesium intake could be essential, particularly in populations at risk, such as older adults and individuals with chronic illnesses or sedentary lifestyles. Practitioners should proactively assess Mg status in patients, especially in at-risk populations. Both hypomagnesemia and hypermagnesemia should be monitored. Severe hypermagnesemia can be fatal and may lead to muscle flaccid paralysis, a decreased breathing rate, pronounced hypotension, and bradycardia.Therapeutic use of Mg supplementation: given the evidence of improved muscle mass, strength, and recovery in certain populations (e.g., patients with cystic fibrosis, chronic obstructive pulmonary disease, and sarcopenia), Mg supplementation should be considered as a therapeutic adjunct in patients experiencing muscle weakness or atrophy.Integrating Mg with other pharmacological and non-pharmacological approaches: Mg could augment the efficacy of existing interventions for muscle atrophy, such as exercise programs, especially in sarcopenic patients. Combining Mg supplementation with resistance training could have synergetic effects on muscle mass and strength.Customized supplement dosages: The optimal dosage and form of Mg should be tailored to individual patient needs, considering factors such as serum Mg levels, muscle health status, and comorbidities. Mg monitoring after supplementation can help in adjusting dosages and assessing efficacy.Fill the evidence gap on Mg supplementation in NMDs: Research suggests that Mg’s role in ion homeostasis could be particularly relevant in NMDs like DMD and others characterized by muscle degeneration and oxidative stress. Mg supplementation could enhance the regenerative capacity of muscle fibers, potentially addressing muscle damage in conditions like myopathies and dystrophies. Future studies should explore Mg supplementation as a possible adjunctive treatment to improve muscle function or slow disease progression in these patients.

### 4.7. A Critical Analysis of Our Findings

In conclusion, we would like to highlight the significant limitations associated with this type of paper and the topic investigated. Scoping reviews inherently lack quality assessments and may be influenced by biases in some of the studies included. Moreover, there is considerable variability in the study designs, populations, and interventions, which complicates the comparison of the results. While we acknowledge that a scoping review cannot replace a more rigorous systematic review when detailed evidence synthesis is required, this paper has identified a knowledge gap on magnesium supplementation in NMDs and has implications for future research.

When assessing the limitations of the current body of evidence, it is crucial to recognize that the majority of preclinical research has concentrated on the effects of dietary magnesium (Mg) deficiency or low Mg levels in vitro, which may not accurately reflect the complex physiological interactions that occur in vivo. This gap limits the applicability of findings to human health outcomes and diminishes the relevance of these studies when considering the broader implications of Mg supplementation in clinical scenarios. Moreover, it may be skewed towards short- to medium-term outcomes, rather than the long-term effects of Mg supplementation on muscle health. The lack of longitudinal studies exploring the lasting impacts of this supplementation makes it challenging to assess the sustainability of any positive findings or to identify potential adverse effects associated with prolonged Mg intake, especially in vulnerable populations. Moreover, the lack of study targeting Mg in the context of NMD leads to variability in the results and conclusions drawn. This inconsistency underscores the need for careful consideration when analyzing existing research and emphasizes the challenges in deriving definitive conclusions about the role of Mg in NMD from diverse and potentially unrelated studies.

Lastly, while we acknowledge that NMDs share some pathogenic mechanisms in common with Mg depletion, there has been insufficient investigation into how Mg affects the specific cellular and molecular processes relevant to NMDs. This research gap limits our understanding of whether Mg supplementation can influence the progression of these diseases or exert distinct effects on muscle repair mechanisms.

## 5. Conclusions

Our findings suggest the beneficial role of an adequate intake of Mg for musculoskeletal health in terms of muscle mass, power, and performance. Moreover, this electrolyte seems to have the potential to improve muscular stem cells and counteract muscle atrophy, supporting its role as a promising therapeutic strategy against sarcopenia and age-related diseases. However, no evidence regarding Mg supplementation in NMDs was found, suggesting that this topic is a potentially intriguing field for further research.

## Figures and Tables

**Figure 1 ijms-25-11220-f001:**
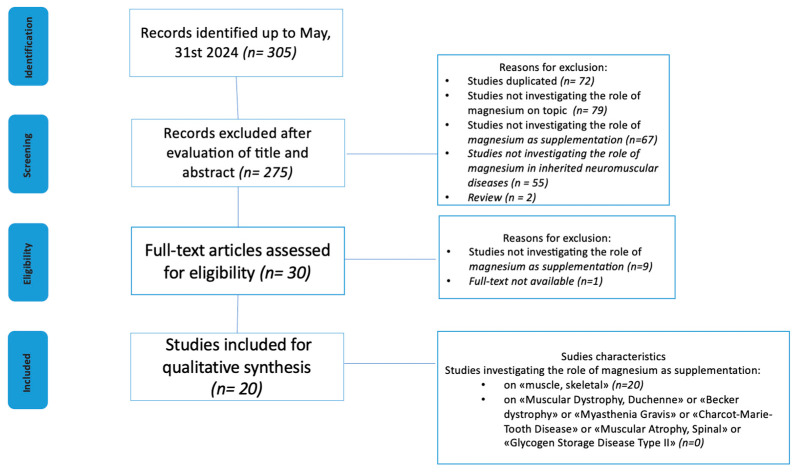
Flow diagram of source selection process.

**Figure 2 ijms-25-11220-f002:**
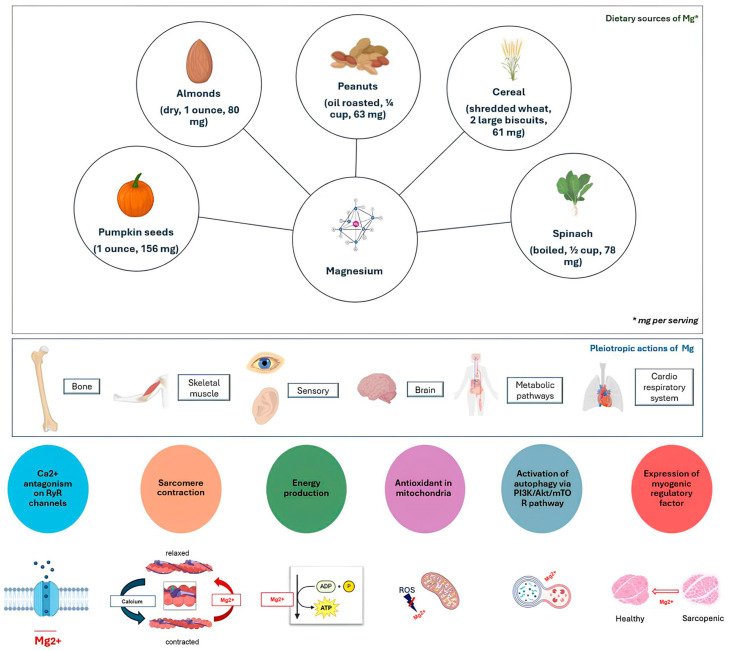
Primary dietary sources of magnesium and its pleiotropic actions, focusing on skeletal muscle health.

**Table 1 ijms-25-11220-t001:** Search strategy.

(“Magnesium” OR “Magnesium Compounds” [Mesh] AND “Muscular Dystrophy, Duchenne” [Mesh] OR “Becker dystrophy” [Mesh] OR “Myasthenia Gravis” [Mesh] OR “Charcot-Marie-Tooth Disease” [Mesh] OR “Muscular Atrophy, Spinal” [Mesh] OR “Glycogen Storage Disease Type II” [Mesh] OR “neuromuscular diseases” [Mesh] OR “Muscle, skeletal”);
2. (“Magnesium” OR “Magnesium Compounds” AND “Muscular Dystrophy, Duchenne” OR “Becker dystrophy” OR “Myasthenia Gravis” OR “Charcot-Marie-Tooth Disease” OR “Muscular Atrophy, Spinal” OR “Glycogen Storage Disease Type II” OR “neuromuscular diseases” OR “Muscle, skeletal”).

**Table 2 ijms-25-11220-t002:** Eligibility criteria.

**Inclusion Criteria**
- English language
- Reference period from inception to 31 May 2024
- Study design: preclinical and clinical studies, including case reports, clinical trials, comparative studies, and observational studies
- Studies investigating the effects of magnesium on skeletal muscle tissue
- Studies including use of magnesium as supplementation for patients with inherited neuromuscular disease
**Exclusion Criteria**
- Books and documents, meta-analyses, reviews, systematic reviews, letters to the editor
- Articles written in other languages.
- Studies investigating use of magnesium in acquired neuromuscular disease
- Studies investigating any kind of use of magnesium not as supplementation and/or treatment for muscle

**Table 3 ijms-25-11220-t003:** Characteristics of the preclinical studies.

Author, Year, Study Design	Sample Size: Total (Group)	Intervention	Outcome	Main Findings
Liu et al., 2007, RCT, in vivo study [21].	n = 96 one-day-old male Arbor Acre broiler chickens divided into low-MgG (n = 48) and CG (n = 48).	Low-MgG had a diet with 1.2 g Mg/kg while CG had a diet with 2.4 g Mg/kg for 6 weeks.	After 6 weeks, measurements of muscular and serum Mg, malondialdehyde (MDA), glutathione (GSH), and mitochondrial electron transport chain (ETC) complex activities were performed.	After 6 weeks of treatment, the low-MgG showed a reduction in muscle and serum Mg, decreased GSH, increased MDA, and ETC complex II and III activity.
Zheng et al., 2021, RCT, in vivo and in vitro study [22].	n= 18 twenty-four-week-old male Sprague–Dawley rats divided into MgG (n = 6) CSG (n = 6) and CG (n = 6).	IN VIVO MgG was treated with lipopolysaccharide (LPS) and corticosteroid (CS)–methyl-prednisolone (MPS) with daily 50 mg/kg Mg oral supplementation; CSG was treated with LPS and CS–MPS; CG was treated as normal control. IN VITRO After 2 days of incubation with differentiation medium, C2C12 myoblast cell line was treated with CS with or without 10 mM Mg chloride for 2 days (MgG-in vitro). The CG-in vitro was treated with solvent only.	After 6 weeks of treatment, serum Mg measurement, tissue composition determination from DXA, functional testing, and histology (muscle fiber size investigation, intramuscular fat infiltration, and fiber typing) of the extensor digitorum longus (EDL) were performed. IN VITRO C2C12 myoblast cell line as myotube atrophy model was used to study the in vitro effect associated with in vivo muscle atrophy.	In MgG vs. CSG, good body composition changes, significant improvements in muscle performance, and an improvement in muscle fiber characterisitcs and cross-sectional area (CSA) were observed. In C2C12 myoblast cultures, MgG promoted larger myotube diameters, indicating enhanced myogenic activity. Both in vitro and in vivo studies showed lower mRNA expression levels of MuRF1 and MAFbx (markers associated with muscle atrophy) in the MgG group, suggesting a protective effect against muscle wasting.
Liu et al., 2021, non-RCT, in vitro and in vivo study [23].	C2C12 myoblast culture; fluorescence-activated cell sorting (FACS) from hind-limb muscles of 24-month-old male mice.	**Part 1** Two-day-differentiated C2C12 myotubes were treated with different concentrations of MgCl_2_ or MgSO_4_ for 96–144 h versus CG treated with 0.8 mM Mg. **Part 2** Six-day-differentiated C2C12 myotubes were subjected to 48 h treatment with 10 nM rapamycin versus dimethyl sulfoxide (DMSO) control group. **Part 3** Isolation of MuSC by FACS treated with 2.5 mM of Mg. **Part 4** Aged muscle regeneration in mice in an NTX-induced muscle injury aged model after (1) intraperitoneal injection of MgSO_4_ or (2) administration of 50 mg/kg/day MgSO_4_ vs. control group MgSO_4_.	The effect of Mg on C2C12 myoblast differentiation and myotube growth on MuSC differentiation and skeletal muscle regeneration after injury was investigated.	**Part 1** At a concentration of 2.5 mM of Mg, MgG vs. CG showed a higher expression of myogenic regulatory factor MyoD, myogenin, and muscle structural protein MyHC. **Part 2** At a concentration of 2.5 mM of Mg, rapamycin suppressed the protein expression of myogenin and MyHC, reduced the fusion index, decreased the myotube diameter, and downregulated protein synthesis. **Part 3** At a concentration of 2.5 mM of Mg, FACS showed - Elevation of the proportion of myogenin + cells and the fusion index; - Increased the diameter of primary aged myotubes. **Part 4** (1) Administration of MgSO_4_ via intraperitoneal injection in mice enhanced myoblast differentiation and increased myotube diameter. (2) Mice treated with 50 mg/kg/day MgSO_4_ showed improvement in regenerative myogenesis, including greater muscle weight, larger myofiber CSA, and enhanced functional outcomes such as increased grip strength and grid-hanging time without significant changes in overall body weight.
Cui et al., 2022, RCT, in vivo and in vitro study [24].	IN VIVO n = 60 male SAMP8 mice randomized into MgG (n = 15), LMHFV(VibG) (n = 15), CTG (Mg + VIB) (n = 15), and CG (n = 15). IN VITRO C2C12 myoblasts.	IN VIVO MgG received 200 mg/kg/day Mg, 5 days/week; VibG had LMHFV (35 Hz, 0.3 g, 20 min/day and 5 days/week); CTG received Mg + LMHFV; CG had LMHFV with the platform powered off. IN VITRO C2C12 myoblasts in 10 groups: (1) CG; (2) VibG; (3) MgG; (4) CTG; (5) VibG + Rapamycin (Ra); (6) VibG + LY294002 (LY); (7) MgG + Ra; (8) MgG + LY; (9) CTG + Ra; (10) CTG + LY. Ra and Ly are inhibitors of PI3K/Akt and mTOR.	IN VIVO After 2, 3, and 4 months SAMP8 mice were evaluated for functional and structural outcomes, serum Mg, EDL for RNA extraction, and quantitative real-time PCR, and tibialis anterior (TA) was investigates with Western blot analysis. At month 0, 2, 3 and 4 post-treatment before euthanasia, whole-body muscle mass (WBMM) and appendicular muscle mass (AMM) were measured in SAMP8 mice through DXA. IN VITRO In C2C12 myoblasts, myotube metrics were quantified (myotube diameter and myotube nuclei numbers), and RNA extraction and quantitative real-time PCR, protein expression with Western blot analysis, and immunohistochemical and immunofluorescence staining of myofibers and C2C12 myotubes (including inflammation markers such as CD206-positive M2 macrophage population) were performed.	IN VIVO In the MgG, at month 4, higher absolute lean mass and percentage lean mass were observed compared to the CG (*p* < 0.05), as well as increased CSA of type IIb muscle fibers compared to all groups (*p* < 0.05), lower proportions of type I and IIa fibers with a higher proportion of type IIb fibers compared to all groups (*p* < 0.001), an enhanced expression of MyoD, MyoG, Myf5, and Myf6 mRNA compared to the CG, and increased mTOR expression relative to the CG and CTG. IN VITRO In comparison to the CG, both VibG and MgG treatments resulted in significant increases in myotube diameter, nuclei count, myogenic index (MI), and the expression levels of Myf5 and Myf6 mRNA (*p* < 0.05).
Takagi et al., 2023, non-RCT, in vivo study [25].	35 four-week-old male Wistar rats divided into 5 groups, each containing n= 7 rats.	Four American Institute of Nutrition (AIN)-93G-based diets with various Mg concentrations for 12 weeks were established: the Mg100 diet (control diet) and the Mg50, Mg25, and Mg8 diets containing 50%, 25%, and 8% of Mg100.	Mg and Ca concentrations were measured and metabolomic analysis was performed in the soleus and gastrocnemius muscles. Moreover, relative expression levels of Renin1 and Fst were evaluated.	Severe Mg restriction (Mg8) led to an increase in muscular levels of 3-phosphoglyceric acid, A decrease in the levels of glucose 6-phosphate, 2-phosphoglyceric acid, phosphoenolpyruvic acid, and fructose 6-phosphate (in soleus and gastrocnemius), an increase in pyruvic acid content (in soleus), a decrease in carnosine and its constituent β-alanine, and an increase in the levels of purine derivatives such as xanthine and uric acid (in gastrocnemius) were measured.
Zocchi et al., 2023, non-RCT, in vitro study [26].	C2C12 murine myoblasts.	Myotubes obtained after 144 h of differentiation from C2C12 murine myoblasts were exposed for 4 days to DM containing severely low (0.1 mM), mildly low (0.5 mM), and physiological (1 mM) MgSO_4_ concentrations.	An analysis of different MyHC isoforms by Western blot of myotubes with an optical microscope and by immunofluorescence using antibodies against the contractile protein MyHC, lipid droplet staining, triglyceride and intracellular lactate quantification, autophagic flux, ROS, and fatty acid oxidation (FAO) analysis was performed.	A significant reduction in thickness and fusion index values (nuclei in myotubes vs. total nuclei) was found in the lower-Mg groups. In severely low Mg conditions, a reduction in MyHC, Myog, and Myomixer levels and the amounts of pAkt and lactate production was observed, along with a substantial downregulation in the levels of MyHC II with no significant differences in total or mitochondrial-generated ROS and a significant increase in NO. A reduction in the amounts of insulin-responsive glucose transporter GLUT4, lower levels of TGs, a decrease in the ratio LC3-BII/BI, and a reduction in the autophagic flux were found in myotubes cultured in low Mg vs. physiological Mg concentrations.

Abbreviations: Randomized Controlled Trial (RCT); Magnesium Group (MgG); control group (CG); corticosteroid group (CSG); muscle RING finger 1 (MuRF1); muscle atrophy F-box (MAFbx); hour (h); muscle stem cell (MuSC); fluorescence-activated cell sorting (FACS); notexin (NTX); myosin heavy chain (MyHC); male senescence-accelerated mouse prone 8 (SAMP8); vibration group (VibG); low-magnitude, high-frequency vibration (LMHFV); combined treatment group (CTG); extensor digitorum longus (EDL); tibialis anterior (TA); dual-energy x-ray absorptiometry (DXA); rapamycin (Ra); LY294002 (LY); whole-body muscle mass (WBMM); appendicular muscle mass (AMM); lean mass (LM); appendicular lean mass (ALM); cross-sectional area (CSA); follistatin (FST); differentiation medium (DM); tryglyceride (TG); radical oxidative species (ROS); myogenin (Myog); nitric oxide (NO); glucose transporter type 4 (GLUT4).

**Table 4 ijms-25-11220-t004:** Characteristics of the clinical studies.

Author, Year, Study Design	Sample Size: Total (Group)	Intervention	Outcome	Main Findings
Clauw et al., 1993, case-report [27].	Male, 43 years old, eosinophilia–myalgia syndrome (EMS).	Twice-weekly intramuscular injections of 1 g of Mg sulfate for 8 weeks (T0–T1). The same protocol was repeated after 12 weeks (T2–T3).	Mg concentration, a self-report questionnaire (including visual analog scales to assess the degree of muscle weakness, pain, spasm, and fatigue), and a magnetic resonance spectroscopy (MRS) evaluation at any time point were collected.	At T1 and T3, an inverse relationship between the severity of symptoms and intramuscular Mg concentration was observed at MRS.
Weller et al., 1998, double-blind, placebo-controlled study [28].	20 athletes with low–normal Mg serum (0.8 mmol·L^−1^) divided into 2 groups (MgG n = 10; PG n = 10).	MgG: daily “Magnetrans forte” (250 mg × 2 Mg oxide). PG: placebo capsules	After 3 weeks, Mg concentration in serum and in skeletal muscle measured by in vivo 31P nuclear magnetic resonance (NMR), exercise tests, “muscle score” derived from a questionnaire on muscle- and performance-related symptoms, and evaluation of neuromuscular activity by EMG was measured.	After 3 weeks, in both groups, no effects of supplementation on any outcomes measured were observed, except for a decrease in the EMG score with no increase in Mg concentration in serum or any cellular compartment studied (except for renal Mg clearance, increased in the MgG).
Zorbas et al., 1999, RCT [29].	40 male athletes divided into four groups (n = 10 in each group): two groups treated (SACS, SHKS) and two untreated (UACS, UHKS).	SACS and SHKS: daily 23 mg Mg lactate/kg for 364 days; UACS and UHKS: No supplementation. The SHKS and UHKS groups were maintained under an average running distance of 1.7 km/day. The SACS and UACS groups experienced no changes in their training.	Mg balance, anthropometric parameters, urinary and fecal excretion, and serum concentration of Mg were collected.	The hypokinetic groups showed a negative Mg balance, a decrease in body weight, body fat, and peak oxygen, and an increase in urinary and fecal excretion and serum concentration of Mg compared to ambulatory groups. No significant differences were observed between the SHKS and UHKS groups for anthropometric and peak oxygen uptake, serum and urinary Mg, faecal magnesium, or Mg balance.
Wary et al. 1999, RCT [31].	30 healthy male volunteers divided into MgG (n = 15) and PG (n = 15).	MgG: three Mg tablets, twice daily (470 mg of Mg lactate and 5 mg pyridoxine); PG: three placebo tablets, twice daily.	Total Mg, lysed erythrocytes Mg, 24 h urine Mg, ionized magnesium in plasma, and intracellular free Mg in skeletal muscle and the brain were measured at baseline and after 28–35 days of treatment.	24 h urine Mg changed significantly for MgG compared to PG.
Aagaard et al., 2005, RCT [32].	59 patients with alcoholic liver disease divided into MgG (n = 25) and PG (n = 34).	MgG—Phase 1 (day 1–2): 30 mmol MgSO_4_ dispensed in 1 L of glucose solution 55 g/L (306 mM) ev; Phase 2 (day 3–week 6): a daily dose of 12.5 mmol Mg oxide per os. PG—Phase 1: 1 L of glucose solution at 55 g/L (306 mM); Phase 2: oral mixture of lactulose, gelatine, and stearate.	Muscle Mg by biopsy from the lateral vastus muscle, muscle strength (evaluated with an isokinetic dynamometer), and muscle mass (determined from two 24 h urinary creatinine excretions) were measured.	After 6 weeks, no significant difference for muscle Mg, muscle strength, and muscle mass between the groups was observed. Compared to baseline, muscle Mg > 7% was observed in the MgG, while muscle strength and muscle mass significantly increased in both groups.
Gontijo-Amaral et al., 2012, double-blind, RCT cross-over study [33].	44 (CF) children and adolescents with cystic fibrosis divided into MgG (n = 22) and PG (n = 22).	Phase 1 (8 weeks): MgG took 300 mg/die of oral magnesium–glycine, while PG took placebo tablets. After 4 weeks of washout, patients changed treatments.	At baseline and at the end of the trial (20 weeks), urinary magnesium (MgU), maximal inspiratory pressure (MIP), maximal expiratory pressure (MEP), and Shwachman–Kulczycki (SK) score were collected.	At the end of the trial, MgU MIP, MEP, and SK score significantly increased in MgG compared with PG.
Zorbas et al., 2010, RCT [30].	40 healthy trained men divided into four groups (n = 10 in each group): two groups treated (SCSs, SESs) and two untreated (UCSs, UESs).	SCSs: daily administration of 3.0 mmol Mg chloride/kg of body weight; SESs: same supplementation + hypokinesia (HK, training with an average distance of ≤2.3 km/d for 364 d); UCSs: no supplementation + no HK; UESs: no supplementation + HK.	Pre-experimental phase of 30 days for all the subjects with no HK training. Experimental phase of 364 days in SCSs (Mg no HK), SESs (Mg, + HK), UESs (no MG + HK), and UCSs (no Mg + no HK).	Compared with the pre-experimental values and the control groups, hypokinetic groups showed a significant decrease in muscle Mg content and a significant increase in plasma, fecal, and urinary Mg.
Córdova Martinez et al., 2017, RCT [34].	24 men divided into basketball players (PB) (n = 12) and CG (n = 12).	PB received 400 mg/day of lactate Mg + standardized diet + two daily training sessions * (a morning session that consisted of a 2 h gym workout and an afternoon session of 3 h of basketball practice). CG received no treatment.	Both groups had blood samples ** taken four times, each separated by 8 weeks (T1: October, T2: December, T3: March, and T4: April).	At T1, between PB and CG, no difference for serum Mg concentrations was observed. At T3 versus T1 and T2, PB showed a significant decrease in serum Mg concentrations, while at T4 vs. T3, PB had higher serum Mg concentrations. During the entire season, in PB, the levels of muscle damage parameters remained the same, except for creatinine ***.
Steward et al., 2019, double-blind placebo-controlled cross-over study [35].	Nine healthy male runners divided into MgG and PG (number not specified).	From day 1 to 7, MgG received 500 mg/day of Mg (MyVitamins^TM^) ****, while PG received a capsule of cornflour. At day 7, both groups performed a 10 km downhill running time trial (TT). On day 9 to 22, both groups had a washout period. From day 23 to 30, the groups performed a cross-treatment/evaluation.	On day 1 and day 8 and day 22 and 30, both groups performed a dynamometer assessment of maximal force production of the knee extensor and flexor muscles (peak concentric knee estensor force (PCKEF); peak concentric knee flexor torque (PCKFT); and peak eccentric knee flexor torque (PEKFT). On days 8–9 and days 29–30 (pre- and post-TT and 1 h and 24 h post-TT), blood samples for IL-6, CK, glucose, and lactate were collected from both groups. On days 8–10 and days 29–31 (pre- and post-TT and 1 h, 24 h, 48 h, and 72 h post-TT), perceived muscle soreness VAS was measured in both groups.	In MgG, at each time point, lower IL-6 and higher IL-6R were observed, while at 24 h, 48 h, and 72 h post-TT less muscle soreness was reported. Both groups reported, immediately post-TT and at 1 h post-TT, higher IL-6 and muscle soreness; this parameter was higher also at 24, 48, and 72 h post-TT. Finally, both groups, at 24 post-TT, showed lower PCKEF, lower PCKFT, and lower PEKFT.
Córdova et al., 2019, non-RCT [36].	18 male professional cyclists divided into MgG (n = 9) and CG (n = 9).	MgG received 400 mg/day of magnesium during 21-day cycling stage race (exceeding RDA), while CG received no supplementation.	Before the race (T1), mid-competition (T2), and before the last stage (T3), ablood sample was collected (serum and e-Mg, CK, LDH, AST, ALT, ALD, Mb, TP, C, Cr).	At T1, both groups showed similar serum Mg and e-Mg levels. At T2–3, both groups decreased significantly in terms of serum Mg and e-Mg levels (significantly more pronounced in the CG). At T2 and T3, the CG had higher Mb values (vs. MgG), while at T1 vs. T3, the MgG showed a negative correlation between Mg and Mb and CK.
Ahmadi et al., 2020, single-blind RCT [37].	44 males aged 50–70 years with moderate-to-severe COPD divided into MgG (n = 23) and CG (n = 21).	MgG: 250 mL of whey beverage fortified with magnesium and vitamin C + dietary advice and routine care. CG: dietary advice and routine care.	At the baseline (T0) and after 8 weeks (T1), blood sample for inflammatory cytokines (IL-6 and TNFα), GSH, and MDA concentrations, muscle parameters (FFM, FFMI, HGS), and HR-QoL with St. George’s respiratory questionnaire (SGRQ) were collected.	In MgG compared to CG, at T1, lower IL-6 levels, higher FFM, FFMI, and HGS values, and a lower score from SGRQ were observed.
Rehafee et al., 2022, RCT [38].	180 patients with orofacial pain and trigger points in the masseter muscle divided into MgG (n = 90) and PG (n = 90).	MgG received an injection of 2 mL of MgSO_4_, while CG received an injection of saline solution.	Pre-injection and 1, 3, and 6 months after injection: pain intensity, MMO, and QoL, through the OHIP-14, were assessed.	At all follow-ups, PG reported a higher VAS. At all follow-ups, in the MgG, a higher MMO (up to 3 months,) and a higher OHIP-14 were reported.
Rondanelli et al., 2024, RCT [39].	59 sarcopenic adults (16 M; 43 F) divided into MgG (n = 30) and PG (n = 29).	MgG received supplementation twice daily of calcium hydroxymethylbutyrate 1500 mg, L-carnosine 125 mg, Lactoferrin 50 mg, sodium butyrate 250 mg, and magnesium 150 mg. PG received supplementation twice daily of isocaloric placebo with the same flavor.	At T0 and T1 (4 months), nutritional assessment, biochemical parameters, anthropometric measurements, body composition, muscle strength, and physical performance were measured.	Compared to placebo, in the MgG, the HGS, chair test, short physical performance battery test, and walking speed test were significantly improved.
Wang et al., 2024, cross-sectional study [40].	10,279 hypertensive adults aged 20 years or older.	Mg intake from diet and supplements assessed using 24 h diet recalls was recorded.	Muscle mass was evaluated by ASMI measured by dual-energy X-ray absorptiometry whole-body scans.	Every additional 100 mg/day of dietary Mg was associated with a 0.04 kg/m^2^ higher ASMI.

Abbreviations: magnesium (Mg); electromyography (EMG); Randomized Controlled Trial (RCT); Magnesium Group (MgG); placebo group (PG); unsupplemented ambulatory control subjects (UACSs); unsupplemented hypokinetic subjects (UHKSs); supplemented hypokinetic subjects (SHKSs); supplemented ambulatory control subjects (SACSs); supplemented control subjects (SCSs); unsupplemented experimental subjects (UESs); supplemented experimental subjects (SESs); unsupplemented control subjects (UCSs); control group (CG); inteleukin-6 (IL-6); inteleukin-6 receptor (IL-6R); chronic obstructive pulmonary disease (COPD); tumor necrosis factor (TNFα); Recommended Daily Allowance (RDA); fat-free mass (FFM); handgrip strength (HGS); glutathione (GSH); malondialdehyde (MDA); health-related quality of life (HR-QoL); fat-free mass index (FFMI); maximum mouth opening (MMO); Oral Health Impact Profile questionnaire (OHIP-14); calcium (Ca), creatinine (Cr); creatinine kinase (CK); lactate dehydrogenase (LDH); aspartate transaminase (AST); alanine transaminase (ALT); aldolase (ALD); total protein (TP); total testosterone (TT); free testosterone (FT); cortisol (C); white blood cell (WBC), platelet (PLT), hematocrit (HCT); myoglobin (Mb); erythrocyte Mg (e-Mg); appendicular skeletal muscle mass index (ASMI); confidence interval (CI). * Except the match day (2 per week); ** blood samples: serum Ca, Mg, Cr, U, CK, LDH, AST, ALT; ALD, TP, TT, FT, C, WBC, PLT, HCT, and Mb. *** Which significantly decreased after T2 and then increased significantly at T3 and T4 compared to T2. **** Magnesium oxide, magnesium stearate, and microcrystalline cellulose.

## Data Availability

Data will be available upon reasonable request.

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
