# Peer review of "Role of Magnesium in Skeletal Muscle Health and Neuromuscular Diseases: A Scoping Review"

_ijms, 2024, doi:10.3390/ijms252011220_

Round 1

Reviewer 1 Report (Previous Reviewer 1)

Comments and Suggestions for Authors

Thank you for providing the revised manuscript. After the revisions, the authors have addressed the major issue present in the initial version—the lack of summary and discussion in the review article. In the revised version, the authors have provided a more logical summary and discussion of the biological functions of magnesium ions in muscle tissue. Based on the extensive structural adjustments and the efforts demonstrated in the manuscript, I recommend accepting this manuscript for publication after the minor revision.

Please remove the editing marks from the manuscript before submission (visible deletion marks in the review version file).

Author Response

REVIEWER 1: Thank you for providing the revised manuscript. After the revisions, the authors have addressed the major issue present in the initial version—the lack of summary and discussion in the review article. In the revised version, the authors have provided a more logical summary and discussion of the biological functions of magnesium ions in muscle tissue. Based on the extensive structural adjustments and the efforts demonstrated in the manuscript, I recommend accepting this manuscript for publication after the minor revision.

Please remove the editing marks from the manuscript before submission (visible deletion marks in the review version file).

A: Thank you for your thoughtful feedback and for recognizing the improvements we've made in our revised manuscript. We're glad to hear that the enhancements to the paper have met your expectations.

Reviewer 2 Report (New Reviewer)

Comments and Suggestions for Authors

The scoping review presented by the authors attempts to establish the state of the art on the role of magnesium (as a supplement and of dietary origin) in musculoskeletal health and neuromuscular diseases. Although the search strategy used by the authors was apparently rigorous (PRISMA-ScR; https://doi.org/10.7326/M18-0850), the way in which the search criteria are described does not allow it to be reproduced by third parties and the systematization of the information is not concrete. It is singular that no systematic or narrative reviews have been included on the topic and the research gaps that have been left (https://doi.org/10.4300/JGME-D-22-00621.1). The authors are suggested to consider the following to improve the manuscript´s uniqueness and scientific soundness:

General/Format. A) Do not forget to describe the meaning of each abbreviation the first time it is mentioned. B) Use italics when needed (e.g. In vivo, In vitro, scientific names of muscles, etc.) C) The second version of the manuscript must be reviewed by a native English speaker or by a formal translation agency.

Title. OK.

Abstract. After considering the modifications to the content discussed below, modify accordingly.

Introduction. A) It is suggested to structure all paragraphs effectively (https://purdueglobalwriting.center/how-to-write-an-effective-paragraph/) from this section onwards. B) Highlight the specific contribution and/or lack of information on the topic that will be disentangled.  

Body of text. A) In the "search strategy" section, authors should describe in more detail the criteria and process for selecting articles so that others can reproduce the search (e.g., putting MeSH IDs for each keyword, Boolean operators, peer-reviewed article selection and filtering, etc.). Check the following examples (doi): https://www.sciencedirect.com/science/article/pii/S1279770723000234  , https://link.springer.com/content/pdf/10.1186/s12978-022-01485-9.pdf  , https://www.jstage.jst.go.jp/article/jnsv/68/3/68_189/_pdf/-char/ja. C) It is suggested to add a bibliometric analysis of more than 10 years on the topic if possible (see).

Tables. A) All tables should be formatted according to journal recommendations. B) It is suggested that Tables 3-4 in its current state be included as supplementary material and a new one be reconstructed with the most relevant information and in a concise form.

Figures. A) All should be provided with a higher resolution (>300 dpi). B) Add footnotes to Figures 1-2. B) Authors are advised to generate a word cloud that allows measuring the interactions between the terms "Mg and..." (see example: https://doi.org/10.3390/ijerph19074165)

References. A) The information contained in systematic reviews on the subject should also have been considered, pointing out the differences with these and the uniqueness of this new report. B) It is suggested to review the format of the references again according to the instructions for authors

Comments on the Quality of English Language

Moderate editing is needed

Author Response

REVIEWER 2:

The scoping review presented by the authors attempts to establish the state of the art on the role of magnesium (as a supplement and of dietary origin) in musculoskeletal health and neuromuscular diseases. Although the search strategy used by the authors was apparently rigorous (PRISMA-ScR; https://doi.org/10.7326/M18-0850), the way in which the search criteria are described does not allow it to be reproduced by third parties and the systematization of the information is not concrete. It is singular that no systematic or narrative reviews have been included on the topic and the research gaps that have been left (https://doi.org/10.4300/JGME-D-22-00621.1).

A: Thank you for your thoughtful feedback and suggestions. In reference to your comments, we would like to clarify that we have identified the research teams involved, which include 5 medical doctors  with expertise in clinical research and scoping reviews. The methodology used in this scoping review is consistent with that employed in our previous published scoping reviews (doi:10.3390/medicina58081014;  doi: 10.3390/medicina57111262; doi:10.3390/medicina57111165; doi: 10.3390/medicina57101118; doi:10.1080/17434440.2021.1927704; doi: 10.23736/S1973-9087.21.06581-3; doi: 10.22203/eCM.v041a20; doi: 10.3390/ijms22052693; doi: 10.3390/nu12072144; doi: 10.3390/nu12010268; doi: 10.1007/s12603-016-0823-x; doi: 10.1186/s12891-016-1086-8.)

According to the methodology of a scoping review only primary studies have been included. The main reason for carrying out this scoping review is the research gap about the use of Mg as a supplement in neuromuscular disorders (NMDs).

The authors are suggested to consider the following to improve the manuscript´s uniqueness and scientific soundness:

General/Format. A) Do not forget to describe the meaning of each abbreviation the first time it is mentioned. B) Use italics when needed (e.g. In vivo, In vitro, scientific names of muscles, etc.) C) The second version of the manuscript must be reviewed by a native English speaker or by a formal translation agency.

A: Thank you for your insightful feedback and suggestions. We have incorporated all the necessary abbreviations and applied italics where appropriate. Additionally, we plan to have the manuscript undergo an English revision upon acceptance.

Title. OK.

Abstract. After considering the modifications to the content discussed below, modify accordingly.

Introduction. A) It is suggested to structure all paragraphs effectively (https://purdueglobalwriting.center/how-to-write-an-effective-paragraph/) from this section onwards. B) Highlight the specific contribution and/or lack of information on the topic that will be disentangled. 

A: Thank you. We have reorganized the introduction to make it more concise and emphasized the lack of information on the topic that justify the aim of the scoping review.

Body of text. A) In the "search strategy" section, authors should describe in more detail the criteria and process for selecting articles so that others can reproduce the search (e.g., putting MeSH IDs for each keyword, Boolean operators, peer-reviewed article selection and filtering, etc.). Check the following examples (doi): https://www.sciencedirect.com/science/article/pii/S1279770723000234  , https://link.springer.com/content/pdf/10.1186/s12978-022-01485-9.pdf  , https://www.jstage.jst.go.jp/article/jnsv/68/3/68_189/_pdf/-char/ja. C) It is suggested to add a bibliometric analysis of more than 10 years on the topic if possible (see).

A: Thank you for your feedback. We have provided a detailed overview of the panel of experts. As shown in Table 1, we conducted a search using both MeSH and non-MeSH terms, utilizing the Boolean operators AND/OR. In Table 2, we defined the inclusion and exclusion criteria used for selection, consistent with the methodology applied in our previous articles (doi:10.3390/medicina58081014;  doi: 10.3390/medicina57111262; doi:10.3390/medicina57111165; doi: 10.3390/medicina57101118; doi:10.1080/17434440.2021.1927704; doi: 10.23736/S1973-9087.21.06581-3; doi: 10.22203/eCM.v041a20; doi: 10.3390/ijms22052693; doi: 10.3390/nu12072144; doi: 10.3390/nu12010268; doi: 10.1007/s12603-016-0823-x; doi: 10.1186/s12891-016-1086-8.) The bibliometric analysis was performed from inception (approximately from 1959) to May 31, 2024.  Three contributing authors to the suggested scoping review (https://doi.org/10.1007/s12603-016-0823-x ) are also co-authors of this paper, following the same methodology recommended for a scoping review.

Tables. A) All tables should be formatted according to journal recommendations. B) It is suggested that Tables 3-4 in its current state be included as supplementary material and a new one be reconstructed with the most relevant information and in a concise form.

A: Thank you for your feedback regarding the tables in our manuscript. I would like to clarify that these tables have already undergone significant revisions during the previous major review. This is the second round of revisions, and we are facing a mismatch due to discrepancies between the corrections made earlier and the new format suggested.We appreciate your suggestions and will ensure that any further revisions will align with the established corrections while also addressing your concerns.

Figures. A) All should be provided with a higher resolution (>300 dpi). B) Add footnotes to Figures 1-2. B) Authors are advised to generate a word cloud that allows measuring the interactions between the terms "Mg and..." (see example: https://doi.org/10.3390/ijerph19074165)

A: Thank you for your suggestions. We have provided a higher resolution image and added the required footnotes. Regarding the word cloud, we believe it would not be meaningful to create one based on our findings, as there is a lack of results for all terms except for "Muscle, skeletal."

References. A) The information contained in systematic reviews on the subject should also have been considered, pointing out the differences with these and the uniqueness of this new report. B) It is suggested to review the format of the references again according to the instructions for authors

A: Thank you for your feedback. According to your suggestions, we have included two systematic reviews on magnesium supplementation in athletes (doi: 10.1186/s12967-024-05434-x, doi: 10.1684/mrh.2018.0430; PMID: 29637897). These reviews support our findings and highlight the complex and varied approaches to magnesium management in the athletic population.

Reviewer 3 Report (New Reviewer)

Comments and Suggestions for Authors

Manuscript ID: ijms-3221924 entitled: Role of Magnesium in skeletal muscle health and neuromuscular diseases: a scoping review

By Sara Liguori, Antimo Moretti, Marco Paoletta, Francesca Gimigliano and Giovanni Iolascon

 The review could be important but there is no new relevant unknow data.

Title

delete word “scoping”, review means comprehensive, scoping, but in terms of reference “Annals of Internal Medicine, Volume 169, Number 7, https://doi.org/10.7326/M18-085, it sounds OK

Keywords

magnesium; dietary supplement; micronutrient overlap, all three words are exaggerated

Introduction

L 36/37 The important part starts with “third major component of bone“ all pevious text is general knowledge

Ref 16 is not clear where it belongs to

Reference Ann Intern Med 2018 Oct 2;169(7):467-473. doi: 10.7326/M18-0850 is missing in this paragraph before other description

Materials and Methods

Search strategy - it is not clear how the keywords are selected, there are many synonyms of magnesium-containing species

Exclusion criteria - why the sentence is separated where it belongs “- Studies investigating any kind of use of magnesium not as supplementation and/or treatment on muscle”

There is no description of the method for summarising the results and the risk of bias between studies.

Clarification of the meaning of qualitatively analysed data, like, software used.

Table 3 and Table 4 are not transparent and not readable; all abbreviations should be written out somewhere to improve readability.

There are still errors in the text, which have been corrected but not yet deleted.

It is very difficult to understand the purpose of a study with too much data without emphasising the main benefit.

In data Author, Year, Study Design, Sample Size: Total (Group), Intervention, Outcome, Main Findings, method used before outcome is missing.

Perhaps author, year, study design can be combined in one column

Sample size: Total (Group) can be simplified: male (N=43), eosinophiliamyalgia syndrome, etc.

The main method for the outcome is not obvious as well as advantages and disadvantages.

Please discuss the limitations of this type of summary and, more generally, the limitations of the topic of the paper.

Most of the discussion is already known, and it is difficult to justify that these findings take up so much space in the paper

Figure 2: Mg2+ sources and the role of mechanisms and health are mixed, but none of this is elaborated to the end, just thrown in.

In conclusion, I see absolutely nothing that is not already known from before.

Comments on the Quality of English Language

English must be improved. Long sentences can be simplified. No repetitions necessary.

There are still errors in the text, which have been corrected but not yet deleted.

Author Response

REVIEWER 3:

Manuscript ID: ijms-3221924 entitled: Role of Magnesium in skeletal muscle health and neuromuscular diseases: a scoping review

By Sara Liguori, Antimo Moretti, Marco Paoletta, Francesca Gimigliano and Giovanni Iolascon

 The review could be important but there is no new relevant unknow data.

A: Thank you for your comment. While it may seem that there is no new or relevant data presented, our review aims to highlight the significant gaps in knowledge regarding magnesium supplementation in neuromuscular diseases (NMD). Despite the existing literature, there remains a lack of comprehensive understanding about its efficacy, optimal dosage, and long-term effects in this specific population. By emphasizing these deficiencies, we hope to encourage further research and discussions that could lead to better therapeutic strategies for individuals with NMD.

Title

delete word “scoping”, review means comprehensive, scoping, but in terms of reference “Annals of Internal Medicine, Volume 169, Number 7, https://doi.org/10.7326/M18-085, it sounds OK

A: Thank you for the suggestion. The comment pertains to ensuring clarity and accuracy in terminology and its relation to the referenced materials. We have enhanced the methodology description by emphasizing the steps necessary for scoping, which makes the term in the title more appropriate.

Keywords

magnesium; dietary supplement; micronutrient overlap, all three words are exaggerated

 A: Thank you! We have eliminated “micronutrient” that may seem redundant with respect to magnesium.

Introduction

L 36/37 The important part starts with “third major component of bone“ all pevious text is general knowledge

A: Thank you! We have rephrased the paragraph by removing redundant parts.

Ref 16 is not clear where it belongs to

A: Thank you. The reference 16 (Gragossian A, Bashir K, Bhutta BS, et al. Hypomagnesemia. [Updated 2022 Nov 4]. In: StatPearls [Internet]. Treasure Island (FL): StatPearls Publishing; 2023 Jan-) available on pubmed at https://www.ncbi.nlm.nih.gov/books/NBK500003/ is cited as source to explain the consequences of the hypomagnesemia in the paragraph “it can be complicated by muscle spasms and cramps, dysesthesia, cardio-vascular manifestations, convulsions, cognitive impairment, and, in severe deficiency cases, hypocalcemia or hypokalemia”

Reference Ann Intern Med 2018 Oct 2;169(7):467-473. doi: 10.7326/M18-0850 is missing in this paragraph before other description

A: Thank you for your suggestion! We have added the reference using the correct sequential numbering.

Materials and Methods

Search strategy - it is not clear how the keywords are selected, there are many synonyms of magnesium-containing species

A: Thank you for your feedback and the opportunity to clarify our methodology regarding the terms selected for our study. We chose these specific terms (“Magnesium” and “magnesium compounds”) to ensure a comprehensive search across a wide range of relevant studies. These two terms indeed encompass several subterms on PubMed that are critical for understanding the broader context of our research such as Magnesium Chloride, Magnesium Hydroxide, Magnesium Oxide, Magnesium Silicates, Asbestos, Amosite, Asbestos, Serpentine, Talc, Magnesium Sulfate, Struvite. This approach was designed to enhance the richness of our dataset and ensure that we did not overlook potentially relevant literature that could inform our findings.

Exclusion criteria - why the sentence is separated where it belongs “- Studies investigating any kind of use of magnesium not as supplementation and/or treatment on muscle”

A: Thank you. We are not sure of understand this comment. Could be a referring to a  typo/editing misleading? For clarity we reported the four exclusion criteria:

 - Books and documents, meta-analyses, reviews, systematic reviews, letters to the editor

- Articles written in other languages.

- Studies investigating use of magnesium in acquired neuromuscular disease

- Studies investigating any kind of use of magnesium not as supplementation and/or treatment on muscle

There is no description of the method for summarising the results and the risk of bias between studies. Clarification of the meaning of qualitatively analysed data, like, software used.

A: Thank you for your feedback. For the description of the method for summarizing the results, we have included Figure 1, to enhance readability.

Regarding the potential risk of bias among the studies, we would like to emphasize that, according to the previously cited work by Tricco et al., “A key difference between scoping reviews and systematic reviews is that scoping reviews are generally conducted to provide an overview of the existing evidence, regardless of methodological quality or risk of bias. Consequently, it is common for scoping reviews to include sources of evidence without critical appraisal. ” Therefore, we did not conduct a risk of bias analysis, as it is not a required component of this type of review.

For the method of the analysis, in the paragraph 2.3, "Data Extraction and Quality Assessment," we detailed the authors involved in the study selection and data extraction processes. This was done independently and without the aid of software. We stated, “All data extracted from the included studies were qualitatively analyzed. The study selection and data extraction were conducted independently by two authors (SL and MP), with a third author (AM) consulted in the event of any disagreements.” This methodology is consistent with those applied in our previous scoping reviews (doi:10.3390/medicina58081014;  doi: 10.3390/medicina57111262; doi:10.3390/medicina57111165; doi: 10.3390/medicina57101118; doi:10.1080/17434440.2021.1927704; doi: 10.23736/S1973-9087.21.06581-3; doi: 10.22203/eCM.v041a20; doi: 10.3390/ijms22052693; doi: 10.3390/nu12072144; doi: 10.3390/nu12010268; doi: 10.1007/s12603-016-0823-x; doi: 10.1186/s12891-016-1086-8.)

Table 3 and Table 4 are not transparent and not readable; all abbreviations should be written out

A: Thank you for your feedback regarding the tables in our manuscript. I would like to clarify that these tables have already undergone significant revisions during the previous major review. This is the second round of revisions, and we are facing a mismatch due to discrepancies between the corrections made earlier and the new format suggested. We have taken your suggestions into account and made some small modifications to enhance clarity and alignment with your recommendations. We appreciate your insights and believe these adjustments will further improve the manuscript.

There are still errors in the text, which have been corrected but not yet deleted.

It is very difficult to understand the purpose of a study with too much data without emphasising the main benefit.

A: Thank you for your insightful comments regarding our paper. We appreciate your perspective and would like to clarify the intent of our work. Our paper does not aim to be revolutionary; rather, it seeks to provide a comprehensive state of the art review on the current understanding of magnesium use, specifically in the context of neuromuscular disease (NMD). NMDs are conditions often associated with low levels of magnesium, yet there is a notable lack of formal recommendations or guidelines regarding magnesium supplementation for these patients. In clinical practice, magnesium is commonly administered, but the absence of robust evidence and standardized protocols raises significant concerns. Our objective is to highlight this gap in the existing literature and to propose a clinical scheme that clinicians can follow when considering magnesium supplementation for patients with NMD. We believe that by synthesizing the current knowledge and addressing the inconsistencies in practice, our work can contribute to more informed clinical decisions and ultimately improve patient outcomes.

In data Author, Year, Study Design, Sample Size: Total (Group), Intervention, Outcome, Main Findings, method used before outcome is missing.

Perhaps author, year, study design can be combined in one column

Sample size: Total (Group) can be simplified: male (N=43), eosinophiliamyalgia syndrome, etc.

The main method for the outcome is not obvious as well as advantages and disadvantages.

A: Thank you for your feedback regarding the tables in our manuscript. I would like to clarify that these tables have already undergone significant revisions during the previous major review. This is the second round of revisions, and we are facing a mismatch due to discrepancies between the corrections made earlier and the new format suggested. We have taken your suggestions into account and made some small modifications to enhance clarity and alignment with your recommendations. We appreciate your insights and believe these adjustments will further improve the manuscript.

Please discuss the limitations of this type of summary and, more generally, the limitations of the topic of the paper.

A: Thank you for your insightful feedback. We have indeed reported both the limitations associated with the type of search conducted and the constraints of the body of evidence available.

Most of the discussion is already known, and it is difficult to justify that these findings take up so much space in the paper

A: To enhance clarity and readability, we have organized the main findings of our scoping review by analyzing the specific pathways through which magnesium affects skeletal muscle. We have included both preclinical and clinical results, highlighting existing discrepancies in the administration and evaluation of magnesium, particularly among athletes. Furthermore, we emphasized the potential utility of magnesium in several chronic diseases affecting skeletal muscle, such as sarcopenia and neuromuscular disorders. Despite the biological plausibility of magnesium's benefits, there is currently a lack of recommendations or guidelines regarding its use in these contexts. We acknowledge that some of the discussions may overlap with existing knowledge; however, we believe our paper adds value by synthesizing these findings in a cohesive manner while drawing attention to gaps in current research and clinical practice. By focusing on the pathways and applications of magnesium that are often overlooked or underexplored, we aim to provide a new perspective and stimulate further research in this important area. We appreciate your feedback and hope that this justification clarifies the presence and relevance of our discussion within the manuscript.

Figure 2: Mg2+ sources and the role of mechanisms and health are mixed, but none of this is elaborated to the end, just thrown in.

A: Thank you for your suggestion. In this picture, we have provided a comprehensive overview of the main sources of magnesium as outlined in the introduction, along with a detailed discussion of its pleiotropic effects on muscle, particularly skeletal muscle, which is the primary focus of our paper. For this reason, we opted not to include other aspects to avoid overwhelming the reader with excessive information.

In conclusion, I see absolutely nothing that is not already known from before.

A: Thank you for your insightful comments regarding our paper. We appreciate your perspective and would like to clarify the intent of our work. Our paper does not aim to be revolutionary; rather, it seeks to provide a comprehensive state of the art review on the current understanding of magnesium use, specifically in the context of neuromuscular disease (NMD). NMDs are conditions often associated with low levels of magnesium, yet there is a notable lack of formal recommendations or guidelines regarding magnesium supplementation for these patients. In clinical practice, magnesium is commonly administered, but the absence of robust evidence and standardized protocols raises significant concerns. Our objective is to highlight this gap in the existing literature and to propose a clinical scheme that clinicians can follow when considering magnesium supplementation for patients with NMD. We believe that by synthesizing the current knowledge and addressing the inconsistencies in practice, our work can contribute to more informed clinical decisions and ultimately improve patient outcomes. We hope that this explanation better conveys the purpose of our study and its potential contributions to the field.

Round 2

Reviewer 2 Report (New Reviewer)

Comments and Suggestions for Authors

Thanks for having accepted most of my suggestions. Systematic tables are still very full of unnecessary words, there is still room to reduce text.

Comments on the Quality of English Language

Minor editing is needed

Author Response

Reviewer 2: Thanks for having accepted most of my suggestions. Systematic tables are still very full of unnecessary words, there is still room to reduce text.

A: Thank you for your valuable feedback and for acknowledging our efforts to incorporate your suggestions. We appreciate your insights regarding the tables, and we have made the necessary adjustments to streamline the content by reducing unnecessary text as per your recommendations. We believe these changes enhance the clarity and readability of the tables.

Reviewer 3 Report (New Reviewer)

Comments and Suggestions for Authors

Manuscript ID: ijms-3221924 entitled: Role of Magnesium in skeletal muscle health and neuromuscular diseases: a scoping review

By Sara Liguori, Antimo Moretti, Marco Paoletta, Francesca Gimigliano and Giovanni Iolascon

 The review could be important but there is no new relevant unknow data.

With the track changes it is impossible to judge tables and quality of the improvment.

Major parts are improved in the manuscript according to my previous suggestions.

What is new in Figure 2 is not clear.

Author claims that there were previous major revision in contrast to mine "This is the second round of revisions, and we are facing a mismatch due to discrepancies between the corrections made earlier and the new format suggested.".

Comments on the Quality of English Language

Minor editing of English language required.

Author Response

REVIEWER 3:

Manuscript ID: ijms-3221924 entitled: Role of Magnesium in skeletal muscle health and neuromuscular diseases: a scoping review

By Sara Liguori, Antimo Moretti, Marco Paoletta, Francesca Gimigliano and Giovanni Iolascon

 The review could be important but there is no new relevant unknow data.

A: Thank you for your feedback. While it might appear that our review lacks new data, our objective is to underscore the important knowledge gaps surrounding magnesium supplementation in neuromuscular diseases (NMD). Despite the available literature, there is still insufficient understanding of magnesium's efficacy, optimal dosage, and long-term effects in this population. By emphasizing these deficiencies, we aim to inspire further research and discussions that could improve therapeutic approaches for individuals with NMD. To enhance clarity and readability, we've organized our main findings by exploring the specific pathways through which magnesium impacts skeletal muscle, incorporating both preclinical and clinical data while highlighting discrepancies in its administration and assessment, especially among athletes. We also spotlight magnesium's potential benefits in various chronic conditions affecting skeletal muscle, including sarcopenia and neuromuscular disorders. Although the biological plausibility of magnesium's advantages is acknowledged, there are currently no established recommendations or guidelines for its use in these contexts. While some of our discussions may overlap with existing knowledge, we believe our paper offers value by synthesizing these findings cohesively and drawing attention to the gaps in current research and clinical practice. By focusing on underexplored pathways and applications of magnesium, we hope to provide a fresh perspective and encourage further investigation in this crucial field. We appreciate your comments and trust that this explanation clarifies our discussion's significance within the manuscript.

With the track changes it is impossible to judge tables and quality of the improvment.

A: Thank you for your valuable feedback regarding the readability of our tables. We understand that the track changes made it challenging to assess the quality of our improvements. To enhance readability, we have made significant changes to the tables and have presented them without the track changes in this leter to reviewers. This approach aims to provide a clearer understanding of the modifications we implemented. We appreciate your understanding and are hopeful that the updated tables will facilitate a better evaluation of our work.

Author, Year,  Study Design

Sample Size: Total (Group)

Intervention

Outcome

Main Findings

Liu et al. 2007,  RCT, in vivo study

n= 96 one-day-old male Arbor Acre broiler chickens divided in lowMgG

(n=48) and CG (n=48)

Low MgG had a diet with 1.2 g Mg/kg while CG had a diet with 2.4 g Mg/kg, for 6 weeks

After 6 weeks , measurement of muscular and serum Mg, malondialdehyde (MDA), glutathione (GSH), mitochondrial electron transport chain (ETC) complex activities were performed

After 6 weeks of treatment, the low MgG showed a reduction in muscle and serum Mg, decreased GSH, increased MDA, and ETC complex II and III activity.

Zheng et al,. 2021,  RCT, in vivo and in vitro study

n= 18 twenty-four week-old male Sprague–Dawley rats divided in MgG (n=6) CSG (n=6) and CG (n=6)

IN VIVO

MgG was treated with lipopolysaccharide (LPS) and corticosteroid (CS) - methyl-prednisolone (MPS) with daily 50 mg/kg Mg2+ oral supplementation;

CSG was treated with LPS and CS – MPS; CG was treated as normal control.

IN VITRO

After 2 days of incubation with differentiation medium, C2C12 myoblast cell line  was treated with CS with or without 10 mM Mg chloride for 2 days (MgG-in vitro). The CG-in vitro was treated with solvent only.

After 6 weeks of treatment, serum Mg, tissue composition from DXA, functional test and histology (muscle fiber size, intramuscular fat infiltration and fiber typing) of the extensor digitorum longus (EDL)

were performed.

IN VITRO

C2C12 myoblast cell line as myotube atrophy model was used to study the in vitro effect associated with in vivo muscle atrophy.

In MgG vs CSG, it was observed a good body composition changes, a significant improvements in muscle performance, an improvement of muscle fiber characterisitcs and cross sectional area (CSA).

In C2C12 myoblast cultures, MgG promoted larger myotube diameters, indicating enhanced myogenic activity.

Both in vitro and in vivo studies showed lower mRNA expression levels of MuRF1 and MAFbx (markers associated with muscle atrophy) in the MgG group, suggesting a protective effect against muscle wasting.

Liu et al 2021,  Non-RCT, in vitro and in vivo study

C2C12 myoblasts culture; fluorescence-activated cell sorting (FACS) from hind limb muscles of 24-month-old male mice.

Part 1

Two-day-

differentiated C2C12 myotubes were treated with different concentrations

of MgCl2 or MgSO4 for 96–144 h versus CG

treated with 0.8 mM Mg2+

Part 2

Six-day-differentiated C2C12 myotubes were subjected to 48

h-treatment with 10 nM rapamycin versus dimethyl sulfoxide (DMSO) control group.

Part 3

Isolation of MuSC by FACS treated with 2.5 mM of Mg

Part 4

Aged muscle regeneration in mice in an NTX- induced muscle injury aged model

after: 1) intraperitoneal injection of MgSO4

2)administration of 50 mg/kg/day MgSO4 vs control group

MgSO4  

It was evaluated the effect of Mg on C2C12 myoblast differentiation and myotube growth, on MuSC differentiation and on skeletal muscle regeneration after injury

Part 1

At a concentration of 2.5 mM of Mg, MgG vs CG showed a higher expression of myogenic regulatory factor MyoD, myogenin and muscle structural protein MyHC

Part 2

At a concentration of 2.5 mM of Mg, rapamycin suppressed the protein expression of myogenin and MyHC, reduced the fusion index, decreased in myotube diameter and down-regulated the protein synthesis.

Part 3

At a concentration of 2.5 mM of Mg, FACS showed

- elevation of the proportion of myogenin+ cells and the fusion index

-increased the diameter of primary aged myotube

Part 4

1)Administration of MgSO4 via intraperitoneal injection in mice enhanced myoblast differentiation and increased myotube diameter.

2)Mice treated with 50 mg/kg/day MgSO4 showed improvement in regenerative myogenesis, including greater muscle weight, larger myofiber CSA, and enhanced functional outcomes such as increased grip strength and grid-hanging time, without significant changes in overall body weight.

Cui et al. 2022,  RCT, in vivo and in vitro study

IN VIVO

n= 60 male SAMP8 mice randomized in MgG (n=15), LMHFV(VibG) (n=15),  CTG (Mg + VIB) (n=15), and CG (n=15).

IN VITRO  C2C12 myoblasts

IN VIVO

MgG received 200 mg/kg/day Mg, 5 days/week;

VibG had LMHFV (35 Hz, 0.3 g, 20 min/day and 5 days/week);

CTG received Mg + LMHFV

CG had LMHFV with the platform powered off

IN VITRO C2C12 myoblasts in 10 groups:

(1)   CG

(2)   VibG

(3)   MgG

(4)   CTG

(5)   VibG +  Rapamycin (Ra)

(6)   VibG +  LY294002 (LY)

(7)    MgG + Ra

(8)   MgG + LY

(9)   CTG + Ra

(10) CTG + LY

Ra and Ly are inhibitors of PI3K/Akt and mTOR

IN VIVO

After 2, 3 and 4 months

in SAMP8 were evaluated functional and structural outcomes, serum Mg, EDL for RNA extraction and quantitative real-time PCR,  and tibialis anterior (TA) for Western blot analysis.

At month 0, 2, 3 and 4 post treatment before euthanasia in SAMP8 were performed whole-body muscle mass (WBMM) and appendicular muscle mass (AMM) through DXA

IN VITRO

In C2C12 myoblasts were quantified myotube metrics (myotube diameter and myotube nuclei numbers), RNA extraction and quantitative real-time PCR, protein expression with Western blot analysis, Immunohistochemical and immunofluorescence staining of myofibers and C2C12 myotubes (including inflammation markers such as CD206-positive M2 macrophage population)

IN VIVO

In the MgG, at month 4, it was observed higher lean mass and percentage compared to CG (p < 0.05), increased CSA of type IIb muscle fibers compared to all groups (p < 0.05), a lower proportions of type I and IIa fibers with a higher proportion of type IIb fibers compared to all groups (p < 0.001) and an enhanced expression of MyoD, MyoG, Myf5, and Myf6 mRNA compared to CG, and increased mTOR expression relative to CG and CTG.

IN VITRO

In comparison to the CG, both VibG and MgG treatments resulted in significant increases in myotube diameter, nuclei count, myogenic index (MI), and the expression levels of Myf5 and Myf6 mRNA (p < 0.05).

Takagi et al. 2023,  Non-RCT, in vivo study

35, four-week-old, male Wistar rats divided in 5 groups, each containing n= 7 rats.

Four American Institute of

Nutrition (AIN)-93G-based diets  with various Mg concentrations for 12 weeks were established: the Mg100 diet (control diet) and the Mg50, Mg25, and Mg8 diets containing 50%, 25%, and 8% of Mg100.

Mg and Ca concentrations and metabolomic analysis were measured in the soleus and gastrocnemius muscles. Moreover,   relative expression levels of Renin1 and Fst were evaluated

Severe Mg restriction (Mg8) led to an increase in muscular levels of 3-phosphoglyceric acid, 2-phosphoglyceric acid, and phosphoenolpyruvic acid, a decrease in levels of glucose 6-phosphate and fructose 6-phosphate (in soleus and gastrocnemius),an increase in pyruvic acid content (in soleus) and a decrease of carnosine and its constituent β-alanine and an increase in the levels of purine derivatives such as xanthine and uric acid (in gastrocnemius).

Zocchi et al, 2023,  Non-RCT, in vitro study

C2C12 murine myoblasts

Myotubes obtained after 144 h of differentiation from C2C12 murine myoblasts were exposed for 4 days to DM containing severe low (0.1 mM), mild low (0.5 mM) and physiological (1 mM) MgSO4 concentrations.

An analysis of different MyHC isoforms by western blot, of myotubes by optical microscope and by immunofluorescence using antibodies against the contractile protein MyHC, a lipid droplet staining, a triglyceride and intracellular lactate quantification,  an autophagic flux, ROS and fatty acid oxidation (FAO) analysis were performed.

A significant reduction of thickness and fusion index values (nuclei in myotubes vs. total nuclei) were found in lower Mg groups

In severe low Mg conditions it was observed a reduction of MyHC, Myog, Myomixer levels, a decrease of the amounts of pAkt  and lactate production and a substantial down-regulation in the levels of MyHC II with  no significant differences in total or mitochondrial-generated ROS and a significant increase of NO

A reduction of the amounts of insulin-responsive glucose transporter GLUT4, lower levels of TGs, a decrease of the ratio LC3-BII/BI, and a reduction of the autophagic flux were found in myotubes cultured in low Mg vs  physiological Mg concentrations,  

Table 3. Characteristics of the preclinical studies

Abbreviations:Randomized Controlled Trial (RCT); Magnesium Group (MgG); control group (CG); corticosteroid group (CSG); muscle RING finger 1 (MuRF1); muscle atrophy F-box (MAFbx); hour (h); muscle stem cell (MuSC); fluorescence-activated cell sorting (FACS); notexin (NTX); Myosin heavy chain (MyHC); male senescence-accelerated mouse prone 8 (SAMP8); vibration group (VibG) ; low-magnitude high-frequency vibration (LMHFV); combined treatment group (CTG); extensor digitorum longus (EDL); tibialis anterior (TA); Dual-Energy X-ray Absorptiometry (DXA); Rapamycin (Ra); LY294002 (LY); whole-body muscle mass (WBMM); appendicular muscle mass (AMM); lean mass (LM); appendicular lean mass (ALM); cross-sectional area (CSA); follistatin (FST); differentiation medium (DM); Tryglyceride (TG); Radical oxidative species (ROS); myogenin (Myog); Nitric oxide (NO); glucose transporter type 4 (GLUT4).

*FoxO3 (forkhead box O) is a key regulatory protein involved in protein synthesis and muscle atrophy. It promotes autophagy and plays a direct role in the transcriptional regulation of autophagy-related genes.

Table 4. Characteristics of the clinical studies

Author, Year,  Study Design

Sample Size: Total (Group)

Intervention

Outcome

Main Findings

Clauw et al, 1993,  Case-report

Male, 43 years old, eosinophilia- myalgia syndrome (EMS)

Twice-weekly intramuscular injections of 1 g of Mg sulfate for 8 weeks (T0-T1). The same protocol was repeated after 12 weeks (T2-T3).

Mg concentration, a self-report questionnaire (including visual analog scales to assess the degree of muscle weakness, pain, spasm, and fatigue), and a  magnetic resonance spectroscopy (MRS) evaluation at any time-point were collected

At T1 and T3 an inverse relationship between severity of symptoms and intramuscular Mg concentration was observed at MRS.

Weller et al, 1998,  double-blind, placebo-controlled study

20 athletes with low-normal Mg serum (0.8 mmol·L−1) divided in 2 groups (MgG n=10; PG n=10)

MgG: Daily “Magnetrans forte" (250 mg x 2 Mg-oxide)

PG: placebo capsules

After 3 weeks Mg concentration in serum and in skeletal muscle by in vivo 31P nuclear magnetic resonance (NMR), exercise tests, “muscle score” derived from a questionnaire on muscle and performance related symptoms and - evaluation of neuromuscular activity by EMG were measured

After 3 weeks, in both groups were observed no effects of supplementation on all outcomes performed, except for a decrease in EMG score with no increase in Mg concentration in serum or any cellular compartment studied (except for renal Mg clearance, increased in the MgG).

Zorbas et al, 1999, RCT

40 male athletes divided in 4 groups (n=10 in each group), 2 groups treated (SACS, SHKS), 2 untreated (UACS, UHKS)

SACS & SHKS: daily 23 mg Mg lactate/kg for 364 days;

UACS & UHKS:No supplementation

The SHKS and UHKS groups were maintained under an average running distance of 1.7 km/day. The SACS and UACS groups experienced no changes in their training.

Mg balance, anthropometric parameters,  urinary and faecal excretion and serum concentration of Mg were collected.

The  hypokinetic groups showed negative Mg balance, decrease of body weight, body fat, peak oxygen and an increase of urinary and faecal excretion and serum concentration of Mg, compared to ambulatory groups. No significant differences were observed between the SHKS and UHKS group for anthropometric and peak oxygen uptake, serum and urinary Mg, faecal magnesium, Mg balance

Wary et al. 1999, RCT

30 healthy male volunteers divided in MgG (n=15) and  PG (n=15)

MgG:3 Mg tablets, twice daily (470 mg of Mg lactate and 5 mg pyridoxine)

PG:3 placebo tablets, twice daily

Total Mg, lysed erythrocytes Mg, and 24-h urine Mg, ionized magnesium in plasma and intracellular free Mg in skeletal muscle and brain were measured at baseline and after 28-35 days of treatment

24-h urine Mg changed significantly for MgG compared to PG.

Aagaard et al, 2005, RCT

59 patients with alcoholic liver disease divided in MgG (n=25) and PG (n=34)

MgG, Phase 1 (day 1-2): 30 mmol MgSO4 dispensed in 1L of glucose solution 55 g/L (306 mM) ev; Phase 2 (day 3- week 6):  a daily dose of 12.5 mmol of Mg oxide per os

PG: Phase 1: 1 L of glucose solution 55 g/l (306 mM);Phase 2: oral mixture of lactulose, gelatine and stearate

Muscle Mg by biopsy from the lateral vastus muscle,  muscle strength (evaluated with an isokinetic dynamometer), and muscle mass (determined from two 24-h urinary creatinine excretions) were measured

After 6 weeks, no significant difference for muscle Mg, muscle strength and muscle mass between the groups was observed. Compared to baseline, muscle  Mg >7% in MgG, while muscle strength  and muscle mass significantly increased in both groups.

Gontijo-Amaral et al,2012,  double-blind, RCT cross-over study

44  Cystic Fibrosis (CF) children  and adolescent divided in MgG (n=22) and PG (n=22)

Phase 1 (8 weeks) MgG took 300 mg/die of oral magnesium-glycine while PG placebo tablets.

After 4 weeks of wash out patients crossed the treatments.

At baseline and at the end of trial (20 weeks) urinary magnesium (MgU), maximal inspiratory pressure (MIP), maximal expiratory pressure (MEP), Shwachman-Kulczycki (SK) score were collected.

At the end of trial, MgU  MIP, MEP,  SK score significantly increased in MgG compared with PG,

Zorbas et al., 2009,  RCT

40 healthy trained men divided in  4 groups (n=10 in each group); 2 groups treated (SCS, SES) 2 untreated (UCS, UES)

SCS: daily 3.0 mmol of Mg chloride/kg of body weight

SES:same supplementation +  hypokinesia (HK, a training with an average distance of   < 2.3 km/d for 364 d)

UCS:No supplementation + no HK

UES:No supplementation + HK

Pre-experimental phase of 30 days for all the subjects with no HK training. Experimental phase of 364 days in SCS (Mg no HK), SES (Mg, + HK), UES (no MG + HK), UCS (no Mg + no HK).

Compared with the pre-experimental values and the control groups, hypokinetic groups showed a significant decrease in muscle Mg content and a significant increase in plasma, fecal and urinary Mg.

Córdova Martinez et al, 2017,  RCT

24 men divided in basketball players (PB) (n=12) and CG (n=12)

PB  received: 400 mg/day of lactate Mg + standardized diet + 2 daily training session *(a morning session that consisted of a 2-hour gym workout and an afternoon session of 3- hour basketball practice)

CG received no treatment

Both groups performed blood samples** taken four times, each separated by 8 weeks (T1: October, T2: December, T3: March, and T4: April).

At T1, between PB and CG no difference for serum Mg concentrations was observed. At  T3 versus T1 and T2, PB showed significant decrease in serum Mg concentrations while at T4 vs T3,  PB had higher serum Mg concentration.  During the entire season, in PB levels of muscle damage parameters remained the same, except for creatinine**

Steward et al., 2019,  double-blind placebo-controlled cross-over study

9 healthy men runners divide in MgG and PG (number not specified)

From day 1 to 7  MgG received 500 mg/day of Mg (MyVitaminsTM)**** while PG received capsule of cornflour. At day 7 both groups performed a 10 km downhill  running  time-trial (TT).

At day 9 to 22 both groups had a washout period. From day 23 to 30 groups performed a cross treatment/evaluation.

Day 1 and day 8, day 22 and 30 both groups performed a dynamometer assessment of maximal force production of the knee extensor and flexor muscles (peak concentric knee estensor force-PCKEF; peak concentric knee flexors torque, PCKFT, peak eccentric knee flexors torque, PEKFT)

At day 8-9 and at day 29-30 (pre and post -TT, and 1 hour and 24 hour post-TT) blood samples for  IL-6, CK, glucose, lactate  were collected from  both groups

At day 8-10 and at day 29-31 (pre and post -TT,1h, 24h, 48h and 72h post-TT) perceived muscle soreness VAS were measured in both groups.

In MgG, at each time, a lower IL-6 and higher IL-6R was observed while at 24h, 48h, 72h post-TT a lower muscle soreness was reported. Both groups reported, immediately post-TT and at 1 h post TT, higher IL-6 and muscle soreness; this parameter was higher also at  at 24, 48, 72h post-TT. Finally, both groups, at 24 post-TT, showed  lower PCKEF,  lower PCKFT  and lower PEKFT.

Córdova et al, 2019,  Non-RCT

18 male professional cyclists divided in MgG (n = 9) and CG (n = 9)

MgG received 400 mg/day of magnesium during  21-day cycling stage race (exceeding RDA) while CG no supplementation

Before the race (T1), mid competition (T2), and before the last stage (T3)
blood sample was collected (serum and e-Mg, CK, LDH, AST, ALT, ALD, Mb, TP, C, Cr)

At T1, both groups showed similar serum Mg and e-Mg levels. At T2-3, both groups decreased significantly serum Mg and e-Mg levels (significantly more pronounced in the CG).  At T2 and T3 CG had higher Mb values (vs MgG) while at T1 vs T3, MgG showed negative correlation between Mg and Mb and CK.

Ahmadi et al., 2020,  single-blind RCT

44 males, aged 50-70 years with moderate to severe COPD divided in MgG (n=23) and CG (n=21)

MgG: 250 ml of whey beverage fortified with magnesium and vitamin C + dietary advice and routine care

CG: dietary advice and routine care

At the baseline (T0) and after 8 weeks (T1) blood sample for inflammatory cytokines (IL-6 and TNFα), GSH and MDA concentrations, muscle parameters (FFM, FFMI, HGS),  and HR-QoL with  St. George’s respiratory questionnaire (SGRQ) were collected.

In MgG compared to CG, at T1 lower IL-6 levels, higher FFM, FFMI and HGS values, and lower score at SGRQ were observed.

Rehafee et al., 2022,  RCT

180 patients with orofacial pain and trigger points in the masseter muscle divided in MgG (n = 90) and PG (n = 90)

MgG received an injection of 2 ml of MgSo4 while CG an injection of saline solution

Pre-injection and  1, 3, and 6 months after injection: pain intensity, MMO, QoL through the OHIP-14 were assessed

At all follow-ups, PG reported a higher VAS. At all follow-ups, in the MgG it was reported a higher MMO (up to 3 months,)  and a higher OHIP-14.

Rondanelli et al., 2024,  RCT

59 sarcopenic adults (16 M;43 F) divided in MgG (n = 30) and PG (n = 29)

MgG received a supplementation twice daily of calcium hydroxymethylbutyrate 1500 mg, L-carnosine 125 mg, Lactoferrin 50 mg, Sodium butyrate 250 mg, Magnesium 150 mg

PG received a supplementation twice daily of isocaloric placebo with the same flavor

At T0 and T1 (4 months) nutritional assessment, biochemical parameters, anthropometric measurements, body composition, muscle strength, physical performance were performed

Compared to placebo, in the MgG, HGS, chair test, short physical performance battery test, and walking speed test  were significantly improved.

Wang et al., 2024,  cross-sectional study

10,279 hypertensive adults aged 20 years or older

Mg intake from diet and supplements assessed using 24-hour diet recalls was recorded

Muscle mass was evaluated by ASMI  measured by dual-energy X-ray absorptiometry whole-body scans

Every additional 100 mg/day in dietary Mg was associated with 0.04 kg/m2 higher ASMI.

Abbrevations: Magnesium (Mg); electromyography (EMG); Randomized Controlled Trial (RCT); Magnesium Group (MgG); placebo group (PG); unsupplemented ambulatory control subjects (UACS); unsupplemented hypokinetic subjects (UHKS); supplemented hypokinetic subjects (SHKS); supplemented ambulatory control subjects (SACS); Supplemented control subjects (SCS); unsupplemented experimental subjects (UES); supplemented experimental subjects (SES); unsupplemented control subjects (UCS);  control group (CG); inteleukin-6 (IL-6); inteleukin-6 receptor (IL-6R); chronic obstructive pulmonary disease (COPD); tumor necrosis factor (TNFα); Recommended Daily Allowance (RDA); fat-free mass (FFM); handgrip strength (HGS); glutathione (GSH); malondialdehyde (MDA); health-related quality of life (HR-QoL); fat free mass index (FFMI);  maximum mouth opening (MMO); Oral Health Impact Profile questionnaire (OHIP-14); calcium (Ca), creatinine (Cr); creatinine kinase (CK); lactate dehydrogenase (LDH); aspartate transaminase (AST); alanine transaminase (ALT);  aldolase (ALD); total protein (TP); total testosterone (TT);  free testosterone (FT); cortisol (C); white blood cell (WBC), platelet (PLT), haematocrit (HCT); myoglobin (Mb); erythrocyte Mg (e-Mg); appendicular skeletal muscle mass index (ASMI); confidence interval (CI).

* except the matches day (2 per week)

**blood samples: serum Ca, Mg, Cr, U, CK, LDH, AST, ALT; ALD, TP, TT, FT, C, WBC, PLT, HCT, Mb

***which significantly decreased after T2, and then increased significantly in T3 and T4 compared to T2.

**** magnesium oxide, magnesium stearate, microcrystalline cellulose

Major parts are improved in the manuscript according to my previous suggestions.

A: Thank you for your valuable feedback!

What is new in Figure 2 is not clear.

A:  sorry for the misunderstanding. We didn’t modify the picture in its contents but we only try to explain the logic behind the figure: this picture has the scope to provide a comprehensive overview of the main sources of magnesium as outlined in the introduction, along with a detailed discussion of its pleiotropic effects on muscle, particularly skeletal muscle, which is the primary focus of our paper. For this reason, we opted not to include other aspects to avoid overwhelming the reader with excessive information.

Author claims that there were previous major revision in contrast to mine "This is the second round of revisions, and we are facing a mismatch due to discrepancies between the corrections made earlier and the new format suggested.".

A: Thank you for your valuable feedback regarding the revision process. We appreciate your attention to the changes made in response to previous comments.

To clarify, we acknowledge that the manuscript has undergone significant revisions prior to this round. Our intention in adjusting the format was to enhance clarity and coherence throughout the text, while also addressing the concerns raised in your earlier review. We recognize that this may have led to noticeable discrepancies, and we are committed to ensuring that all revisions align with the overall vision of the paper. We will carefully review the previous corrections in light of your suggestions, ensuring coherence and consistency throughout the manuscript.

Round 3

Reviewer 3 Report (New Reviewer)

Comments and Suggestions for Authors

Authors improved the manuscript according to my  suggestions.

Comments on the Quality of English Language

Minor editing of English language required.

Author Response

Reviewer:

Authors improved the manuscript according to my suggestions.

Comments on the Quality of English Language Minor editing of English language required.

Author: Thank you for your positive feedback and for acknowledging the improvements made to the manuscript. We appreciate your suggestions and have taken them into account to enhance the clarity and quality of our writing.

We have carefully reviewed the manuscript once more for minor editing of the English language to ensure it meets the highest standards

This manuscript is a resubmission of an earlier submission. The following is a list of the peer review reports and author responses from that submission.

Round 1

Reviewer 1 Report

Comments and Suggestions for Authors

The manuscript titled "Role of Magnesium in skeletal muscle health and neuromuscular diseases: a scoping review” emphasized the critical role of Mg in skeletal muscle, particularly in neuromuscular diseases by PRISMA-ScR consisting of pre-clinical and clinical research papers. The literature coverage of this review manuscript is extensive and up to date. I believe that further structural optimization and organization of this manuscript will significantly enhance our understanding of the crucial role of magnesium ions in muscle function and disease treatment.

For this manuscript, I have the following suggestions and comments:

1. A review needs to offer new perspectives or insights, rather than merely summarizing existing literature. The manuscript spends most of its length listing existing research results, but it does not adequately organize, synthesize, or summarize these findings.

2. Review articles should ensure that the structure is logical, the sections are clearly arranged, and the content is coherent. This review manuscript categorizes existing research results into clinical and pre-clinical sections in the main body, which is overly simplistic and lacks organization. Compared to the classification based on pre-clinical and clinical stages, using biological functions and processes as the basis might be better. For example, magnesium ions' effects on muscle differentiation, adipose formation, inflammatory responses, muscle strength, muscle fatigue, etc.

3. The content in Table 3 and Table 4 of the manuscript highly overlaps with Sections 3.1 (Main Findings in Preclinical Studies) and 3.2 (Main Findings in Clinical Studies). The tables do not provide significant additional support and occupy a considerable amount of space.

4. The readability of Tables 3 and 4 is poor. The tables primarily present existing results in a rigid manner and do not standardize the presentation style and format of different studies. By reading the tables, readers still find it difficult to understand the experimental design and conclusions.

For example, but not limited to (the tables do not include row numbers):

1) “Sample size: Total (Group)” column, some sample size indicated by number but some of them indicated by “n=18”

2) “Part 2 2.5 mM MgG For 48 h 10 nM rapamycin DMSOg”, it’s quite difficult to understand the treatment setting.

3) “In an NTX- induced aged muscle injury model”, I suppose it should be “In an NTX- induced muscle injury aged model”

4) “functional a,1 and structuralb outcomes”

5) Magnesium ions are represented as 'Mg' in some places, while in other places they are represented as '[Mg]'. Same cases as “WBMM/percentage”, “ALM%”, “LM%”, “type IIa fibre %”

6) When presenting research results, specific marker genes should be briefly explained rather than simply transferred to this manuscript. For example, CD206, FOXO3, etc.

7) Unclear statement like “higher PAX7-positive cell”.

8) inaccurate statements like “increase expression protein (pAkt)”, it should be the phosphorylation level.

9) Spelling error. “myotuber diameter”, it should be “myotube diameter”

Comments on the Quality of English Language

The quality of English language in the textual description of this manuscript is very good, with no obstacles to reading.

Reviewer 2 Report

Comments and Suggestions for Authors

The authors present a review of the role of magnesium in skeletal muscle health and neuromuscular diseases. The review is well written with a broad review of the literature. Following are some suggestions to improve the readability of the article,

1. The sentence in lines 44-45 has been repeated in lines 48-50. 

2. In table 3 and 4, the outcome and main findings sections could be summarized more concisely.

3.  In table 3, it would be good to separate the in vitro and in vivo studies as two sections.

4. In a review paper, it is not required to have a separate results and discussion sections. It would be good to combine the results and discussion section. It is recommended that the authors make subsections while discussing the literature findings. 

5. The font sizes in figure 2 are too small and the text is difficult to read. Please increase the font sizes and change the colors to make to text more readable. 

Comments on the Quality of English Language

English language quality is fine.